# A Deep Learning Neural Network to Classify Obesity Risk in Portuguese Adolescents Based on Physical Fitness Levels and Body Mass Index Percentiles: Insights for National Health Policies

**DOI:** 10.3390/bs13070522

**Published:** 2023-06-21

**Authors:** Pedro Forte, Samuel Encarnação, António Miguel Monteiro, José Eduardo Teixeira, Soukaina Hattabi, Andrew Sortwell, Luís Branquinho, Bruna Amaro, Tatiana Sampaio, Pedro Flores, Sandra Silva-Santos, Joana Ribeiro, Amanda Batista, Ricardo Ferraz, Filipe Rodrigues

**Affiliations:** 1CI-ISCE, Higher Institute of Educational Sciences of the Douro (ISCE Douro), 4560-708 Penafiel, Portugal; pedromiguel.forte@iscedouro.pt (P.F.); samuel01.encarnacao@gmail.com (S.E.); soukaina.hattabi@issepk.rnu.tn (S.H.); luis.branquinho@iscedouro.pt (L.B.); brunaraquel97@hotmail.com (B.A.); pedro.flores@iscedouro.pt (P.F.); sandra.santos@iscedouro.pt (S.S.-S.); joana.ribeiro@iscedouro.pt (J.R.); amandabatistagrd@yahoo.com.br (A.B.); 2Department of Sport Sciences, Instituto Politécnico de Bragança (IPB), 5300-253 Bragança, Portugal; mmonteiro@ipb.pt (A.M.M.); tatiana_sampaio30@hotmail.com (T.S.); 3Research Centre in Sports Sciences, Health Sciences and Human Development (CIDESD), 5001-801 Vila Real, Portugal; ricardompferraz@gmail.com; 4Department of Pysical Activity and Sport Sciences, Universidad Autónoma de Madrid, Ciudad Universitaria de Cantoblanco, 28049 Madrid, Spain; 5Department of Sport Sciences, Polytechnic Institute of Guarda, 6300-559 Guarda, Portugal; 6High Institute of Sports and Physical Education of Elkef, University of Jendouba, Kef 7100, Tunisia; 7School of Health Sciences and Physiotherapy, University of Notre Dame Australia, Sydney 2007, Australia; sortwellandrew@gmail.com; 8Department of Sports Sciences, University of Beria Interior, 6201-001 Covilhã, Portugal; 9Research Center in Sports Performance, Recreation, Innovation and Technology (SPRINT-IPVC), Polytechnic Institute of Viana do Castelo, 4960-320 Viana do Castelo, Portugal; 10ESECS—Polytechnic of Leiria, 2411-901 Leiria, Portugal; filipe.rodrigues@ipleiria.pt; 11Life Quality Research Center (CIEQV), 2040-413 Leiria, Portugal

**Keywords:** metabolic syndrome, inflammation, immunity, energy expenditure, physical exercise, public health

## Abstract

The increasing prevalence of overweight and obesity among adults is a risk factor for many chronic diseases and death. In addition, obesity among children and adolescents has reached unprecedented levels and studies show that obese children and adolescents are more likely to become obese adults. Therefore, both the prevention and treatment of obesity in adolescents are critical. This study aimed to develop an artificial intelligence (AI) neural network (NNET) model that identifies the risk of obesity in Portuguese adolescents based on their body mass index (BMI) percentiles and levels of physical fitness. Using datasets from the FITescola^®^ project, 654 adolescents aged between 10–19 years old, male: 334 (51%), female: *n* = 320 (49%), age 13.8 ± 2 years old, were selected to participate in a cross-sectional observational study. Physical fitness variables, age, and sex were used to identify the risk of obesity. The NNET had good accuracy (75%) and performance validation through the Receiver Operating Characteristic using the Area Under the Curve (ROC AUC = 64%) in identifying the risk of obesity in Portuguese adolescents based on the BMI percentiles. Correlations of moderate effect size were perceived for aerobic fitness (AF), upper limbs strength (ULS), and sprint time (ST), showing that some physical fitness variables contributed to the obesity risk of the adolescents. Our NNET presented a good accuracy (75%) and was validated with the K-Folds Cross-Validation (K-Folds CV) with good accuracy (71%) and ROC AUC (66%). According to the NNET, there was an increased risk of obesity linked to low physical fitness in Portuguese teenagers.

## 1. Introduction

Worldwide, the prevalence of obesity in adolescents has increased and become a serious problem, exacerbated by restricted physical activity resulting from lockdowns during the COVID-19 pandemic. Consequently, this issue has undesirable effects on the health of young people, the healthcare system, and also the economy [1].

This has become problematic in Portugal, because over the last five years, adolescent obesity rates have been rising in Portugal, as they have in many other nations, placing an increased load on the healthcare system. This is especially due to diverted healthcare investments by the government health authorities, such as toward medications [2]. World data from 161 countries shows that at 2019, obesity impacted the world economy by US$ 2.2 trillion [3]. In Portugal, a prospective population study showed that absenteeism related to obesity was responsible for € 238 million in extra costs per year [4]. In a governmental perspective, prevention is considered an essential measure because it can avoid extra healthcare expenses via medication and non-communicable disease (NCD) control [5]. In 2019, the World Health Assembly give continuity in the world health organization’s (WHO) global action to NCD prevention and control, underlining preventive goals until 2030. Regarding obesity prevention, the WHO consider non-pharmacological treatments such as healthy active lifestyles involving good physical activity levels and dietary behavior modifications as effective with minimal economic impact and side effects, therefore offering a more sustainable approach across the lifespan [6].

Data from a network of health researchers, professionals, and stakeholders from 49 different countries called the “Active Health Kids Global Alliance”, revealed that in Portuguese children and adolescents the categories “Organized Sports Practice and Physical Activity”, “Overall Physical Activity”, and “Physical Fitness” were classified with poor scores [7]. According to numerous meta-analyses, youth obesity risk is highly linked to poor physical fitness [8,9,10]. In addition, obese subjects had a higher inflammatory profile and worsened immune system, which negatively impact all body systems associated with hypertension, cardiovascular disease, type two diabetes, metabolic syndrome, cancer, and autoimmune disease [11,12]. Even though obesity is a multifactorial disease, the predominant risk factor is an imbalance between energy expenditure and energy intake [13]. Physical activity and level of physical exercise are the most modifiable factors of energy expenditure in adolescents. A localized and contextually developed physical exercise program in Portugal such as the Fitescola^®^ program has demonstrated the benefits of lifestyle modification through the lens of physical activity. The program’s target is encouraging young people to lead healthy lifestyles through sports practice [14]. The exercises in the program emphasize several aspects of physical fitness, including muscular strength, flexibility, endurance, and agility. A key characteristic of the program was periodical evaluations that allow increased monitoring, and individual awareness, of health-related physical fitness levels and obesity status (i.e., body mass index: BMI) by percentiles to affirm the benefits of the program [15].

Thus, based in the worrisome evidence about the adolescent’s obesity rates, the Portuguese government, the European Union, and other Portuguese authorities developed the Fitescola^®^ program as an initiative that has been used to evaluate the degree of physical fitness of Portuguese youths, and it raises worries about their health [16]. The Fitescola^®^ brings possibilities to compile datasets with considerable information about the adolescent’s overall physical health [17]. These populational data are very relevant because they possibly have a deep comprehension of the epidemiology; however, producing information that provides direct knowledge to the applied practice is a challenging goal that comes to light in science [18]. In this way, data science is the study field that looks for the comprehension, organization, and analysis of structured or unstructured datasets to produce objective and precise information. Data scientists use several algorithm techniques to extract valuable interpretations from the data [19].

The NNET are deep learning (DL) methods that, in other words, teach the computer software from computers to process data inspired by the biological neural networks present in the human brain [20,21]. The main difference between DL from machine learning (ML) is that in DL the process happens without the supervision of humans and the system can learn and have precise and intelligent decisions by its own way [22]. Indeed, it can learn and identify patterns in complex datasets, providing information for disease diagnosis and prediction, which in turn helps epidemiological modeling, environmental health, healthcare management, and public health surveillance [20,21,23]. Consequently, developing precise and consistent models may be useful for the early management and prevention of obesity-related health disorders.

In the current literature, we identified some studies using ML to predict obesity in North American [24,25,26,27,28], South Korean [29], and Turkish adolescents [30,30]. Although, none these authors considered direct physical fitness levels as potential deterministic variables for their analysis. Similarly, we did not find any previous study which has investigated obesity risk in Portuguese adolescents. In addition, we published preliminary results about the relationship between lower physical fitness and obesity in Portuguese adolescents [31]; however, we have unknown studies performing NNET or similar predictive validation techniques in childhood and adolescence obesity screening at a national level. Different to conventional regression analysis, DL models, when well trained, produce more robust generalization capability, which is very valuable in real life.

Following this, in this paper we aim to develop a NNET to classify the risk of obesity in Portuguese adolescents based on their BMI percentiles and levels of physical fitness. Based on this, our main hypotheses are first, the obesity of the adolescents will correlate inversely with their physical fitness levels; second, the NNET will present good validation performance to learn data patterns; and third, our NNET could be helpful to national health policy making in Portugal as they attempt to combat the obesity epidemic during adolescence.

## 2. Materials and Methods

### 2.1. Study Design

This observational, cross-sectional study aims to classify the obesity risk in Portuguese adolescents from both sexes, based on their BMI percentiles and physical fitness. This study was approved by the Scientific Board of the Higher Institute of Educational Sciences of the Douro (PF: 10.2021). Before data collection, the objectives of this study were explained to all parents or legal guardians and signed informed consent was obtained individually. The parents or legal guardians signed a written informed consent afterward. After approval, all minor participants were asked to sign a written informed consent.

### 2.2. Dataset

Datasets were provided by FITescola^®^, a Portuguese project which aims to promote healthy behavior in children and adolescents [17] that happened in September 2021 on the Paredes council. Thus, in total, a cohort of 654 adolescents aged between 10–19 years old—male: 334 (51%), female: *n* = 320 (49%), age of 13.8 ± 2 years old, height of 162 ± 58 cm, and body weight of 56.5 ± 16 kg, were selected to participate in this study. The obesity risk classifications were determined as the independent variables of the adolescent’s age, sex, aerobic fitness (AF), upper limb strength (ULS), horizontal jump (HJ), 40-m sprint time (ST), and lower limb flexibility (LLF), and the target variable was the obesity risk classification based on BMI percentiles >85th or upper [32]. We described the participant’s characteristics in absolute values, and performed correlation analysis to complete the sample characterization [22]. Table 1 shows the characteristics of the participants.

The eligibility criteria considered adolescents of both sexes, free of any disabling condition, and with ages ranging between 10–19 years old, adopting the WHO classification of 2 adolescence categories (15–19 years old) [33]. Figure 1 below shows the frequency of subjects for each outcome.

Figure 2 shows the frequency of adolescents classified with obesity according to their sex.

### 2.3. Data Collection

In our study, with FITescola^®^, we collected the following data: we measured the obesity status with the adolescent’s BMI percentiles, and their physical fitness (AF with the Yo-Yo test, HJ with the maximal horizontal jump test, ST with the 40-m sprint time test, ULS with the maximal push-up test, and LLF with the sit and reach test) [17]. The BMI measuring procedures and the FITescola^®^ testing protocol are described as follows:

#### 2.3.1. Body Mass Index Percentiles

To obtain the BMI of the adolescents, we followed specific measurement procedures. First, we weighed the subjects while they were barefoot and wearing lightweight clothing. They stood upright and waited for the brand scale’s reading to stabilize. We set a scale with a precision of 100 g. Next, we measured the height of the adolescents. They stood barefoot with their feet together, and their backs pressed against the stadiometer scale. We positioned the stadiometer’s headpiece at the top of the subject’s head, compressing the higher part of the head (vertex). We set a stadiometer with a precision of millimeters. To calculate BMI, we used the body weight divided by the square of the height (in kilograms per square meter). We considered the BMI percentile cutoffs: Underweight: <5th percentile; Healthy Weight: 5th to <85th percentile; Overweight: 85th to <95th percentile; and Obesity: ≥95th percentile. Then, we classified the adolescents with a BMI percentile: ≥85th as under obesity risk [32].

#### 2.3.2. Aerobic Fitness

To measure AF, we applied the Yo-Yo test [17]. We positioned two cones spaced apart 20 m, and we positioned the participant standing behind the starting line. The participant began running back as they heard the automatic sound signaling. Their instruction was to touch the 20-m line before reversing direction and running again to the starting line upon hearing the subsequent sound signal. We set the audio signal to monitor the participant’s speed during the test. We set the first lap at a speed of 8.5 km/h. We increased the participant’s running speed by 0.5 km/h every minute as they reach a maximal 120 rounds. We determined the maximal laps performed as the final text result.

#### 2.3.3. Horizontal Jump

To evaluate the lower limb power of the adolescent, we employed the maximal horizontal jump test [17]. A horizontal line was drawn as the starting point, and reference lines were marked every 10 cm (with the first reference line placed at a distance of 1 m from the starting line). To facilitate the measurement of the distance reached, we placed a measuring tape with an accuracy of 1 mm perpendicular to the horizontal lines. The participant positioned themselves behind the starting line with their feet shoulder-width apart. Starting from a standing position, the participant bent their knees, pulled their arms behind their back making a countermovement, and performed a maximal jump forward. We gave two attempts to capture the best result out of the two evaluations measuring the distance from the starting point to the participant’s heel. We set the measurements in a centimeter scale.

#### 2.3.4. 40-m Sprint Time

We conducted the standardized 40-m sprint test to evaluate the maximal running speed of the adolescents [17]. Prior to the test, the participants performed a 3-min warm-up to activate their muscles and reduce the risk of injuries. We placed two signaling cones, marking the course’s starting and finishing points. The participant positioned themselves behind the starting line in a standing position, with their lower limbs aligned in an anteroposterior direction and their trunk slightly inclined forward. Upon receiving the evaluator’s signal of “prepare, now,” the participant began the sprint, aiming to achieve their highest possible speed. We used a stopwatch set at milliseconds to measure the time taken from the start to the finish line. We conducted two trials, recording the best result.

#### 2.3.5. Upper Limb Strength

We applied the push-up test to assess the adolescent’s upper limb strength [17]. We instructed the participants to assume a plank position with their feet hip-width apart and tiptoes touching the ground. The starting position involved placing the hands directly in line with the shoulders, fingers pointing forward. Throughout the test, we required the participants to aways maintain their trunk in a plank position. The participants started the movement by flexing their elbows at a controlled pace of 1 s for the eccentric phase, lowering the body towards the ground. This movement was followed by a full elbow extension at a pace of 1 s to return to the starting position. The participants repeated the sequence until they reached muscle failure. We recorded the maximum number of repetitions achieved.

#### 2.3.6. Lower Limb Flexibility

We applied the sit and reach test to measure the lower limb flexibility of the participants [17]. We instructed the participants to sit on the floor, barefoot, facing a box. One leg was fully extended, with the foot touching the box, while the other leg was flexed, with the sole of the foot firmly on the floor, aligned with the extended leg’s knee. Once in position, we instructed the participant to flex their torso forward four times, aiming to reach as far as possible on the ruler placed atop the box. We required them to hold the position for one second on the fourth repetition. We positioned the bent knee on the outside of the arms to allow the trunk to advance. The palms of the adolescent’s hands were facing downwards, with the fingers extended and the hands superimposed while maintaining the initial leg position. The objective was to achieve the maximum measurement, we recorded the distance based on the value reached by the middle finger on the ruler. We required the participant to maintain this position for at least one second. The participants repeated the test, but with another leg. We considered the final result as the average of the two measurements in centimeters.

## 3. Results

### 3.1. Convolutional Neural Network Developing

Procedures were performed working in the Python^TM^ programing language [34]. Then, the neural network was developed from the explanatory variables (Age, Sex, AF, ULS, HJ, ST, LLF) and the outcome target (risk of obesity acquired by BMI percentiles). The target variable was identified as dichotomous on the dataset (0 = negative/non-obese, 1 = positive/obese) [35].

Figure 3 shows the correlation matrix between all variables present in the dataset, adopting effect size cutoffs of ≤0.10—small, ≥0.20—moderate, ≥0.30—large, and ≥0.40—very large, according to the assumptions of Funder et al. [36].

Then, we built a three 1-convolutional layers NNET, aiming to classify the obesity risk of the adolescent attributed to their BMI percentiles and physical fitness levels. The NNET’s hyper parameters can be seen in Table 2. The “tensorflow” function was used to perform the training on the NNET model. In addition, the three convolutional layers were activated by the “DENSE” function, which allows every individual neuron to get inputs from all neurons present in the layers, thus, performing a complex learning pattern which provides high accuracy but is characterized in a simple structure. Then, in the first layer, the number of inputs was arranged to 7 (number of independent variables), and the number of units (neurons in the first layer) was 4, determined by the following Equation (1):(1)$$Neurons=\frac{Inputs+outputs\;}{2}$$

Where, is performed by the division of the total number of independent variables (inputs), plus the number of the dependent (target) variable (outputs) per two, then, by determining 4 neurons to the model. The Kernel initializer adopted the function “uniform” intending to provide equalized probabilities over the range of data. Also, duringthe first and second layers, the “relu” activation function was performed to disable all neurons being used at the same time, and in this way, if some input was presented negatively, this function activates only the needed neurons, becoming a more perfected and efficient NNET computation. In the third convolutional layer, we performed the “sigmoid” function since the target variable was identified as 0 or 1, allowing the NNET to learn adequate non-linear relationships between the datasets. In the model compilation for the outcome metrics, the adaptive moment estimation “adam”, and the function “binary_crossentropy” was used, and as the main metric output, the percentage of accuracy was determined for this case. As the performance evaluation method, we selected ROC AUC [37]. Regarding the model of training, 30% of the total dataset was determined to be used in the data training and 70% for the data testing [22]. The model summary is presented in Table 3.

Regarding the fitting model processing, we determined the batch size of 16 units, according to the recommendations of Kandel & Castelli [38], which referred it as an optimizer for the accuracy of the inferences. After the definitive model fitting, we verified an accuracy of 75% in classifying obesity risk, Table 4.

We started the epoch’s number with 24 units (three times the number of columns in the dataset), based on a study by Shahriari et al. [25], which provides information about the larger variability of the accuracy performance over different epochs units. Then, after comparing 24, 50, and 100 epochs, the model’s accuracy did not improve; thus, in a simplified manner, we selected 24 units for the final training model. In addition, the confusion matrix showed an accuracy of 100%, being perfect for identifying all true positives (160), and all the true negatives (*n* = 35), Figure 4. In addition, the ROC AUC analysis presented a satisfactory model performance evaluation (64%) in finding the true positive and negative cases, Figure 5. The full NNET’s programming script can be found in the Appendix A.

### 3.2. NNET Validation

In addition, as a complementary validation method for the NNET, we performed a K-Fold Cross-Validation (K-Fold CV) with the same arrays. K-Fold CV is very well applied when datasets used in NNET have a limited size or require additional validation. Therefore, K-Fold CV splits the data into K subsets, where the most common subsets adopted are 10 K subsets. Each fold is used for data validation, and one folder (K-1) is used to train the data [39]. In our case, we used K-10 subsets, which split the data into 6 folds with 65 rows and 4 folds with 66 rows (Total = 654 rows). We describe the K-Folds CV in Figure 6. As the main results, the K-Fold CV showed 71% accuracy and 66% ROC AUC after validation analysis over all folds. The full programming script of the K-Folds CV can be found in the Appendix A.

## 4. Discussion

This study aimed to develop an NNET model that identifies the obesity risk of Portuguese adolescents based on their BMI percentiles and physical fitness levels. We proved our three hypotheses when we, first, found inverse and moderate size correlations between the adolescent’s physical fitness and their BMI percentiles; second, found that the NNET understood the adolescent’s obesity risk with an accuracy of 75%, ROC AUC = 64%, based on the correlations between their physical fitness and BMI percentiles during the fitting modeling process. Therefore, our NNET was able to find all true positives (*n* =160) and all true negatives (*n* = 35) (100% accuracy score in the confusion matrix analysis for the testing array; and third, we found that^,^ the K-Folds CV validated our NNET when it was presented with an accuracy of 71%, ROC AUC = 66%, which are considered good validation scores, and confirmed our study’s applicability. Despite the high precision of the model to find true and negative cases of obesity (100%), the non-completed comprehension in the fitting model (75%) and the moderate size coefficient correlations between only AF, ULS, and ST with the adolescent’s obesity risk explains that other variables are possibly associated with the obesity incidence in this cohort. The BMI percentiles and categories are explained by several factors such as genetics [40], environment [41], a lifestyle based on diet [42], physical activity [43], comorbidities associated with other diseases [44], medication [45], sleep quality [46], and health problems related to metabolic syndrome [47]. In this study, the authors used only physical fitness variables, age, and sex to identify the obesity risk in the participants [15].

As cited above, the big population study called “Global Matrix 3.0 Physical Activity Report Card”, performed by Aubert et al. [48], counts 49 countries participating in the initiative and reported Portugal as the bests positive grade classification for the school organization: A, whereas presented negative scores in moderate to low grades for overall physical activity: D-, organized sport and physical activity: C-, active play: D+, active transportation: D-, sedentary behavior: C+, physical fitness: C. Our results agree with the results from Aubert et al. [48] by showing similarities in the poor physical fitness levels among the different analysis. On the other hand, our study focused on analyzing obesity from a different prism and making a phenomenological validation based on AI inferences. DL can optimize the data learning based on the raw data properties, optimizing the unsupervised process and, thus, report reliable results in applied practice [49].

We perceived correlations of moderate effect size for AF, ULS, and ST with the BMI percentiles followed by inverse and small size correlations between the AR, HJ, and LLF with the BMI percentiles, showing that in our study, physical fitness levels contributed to the obesity risk of the adolescents. Obesity is marked by chronic inflammation and suppressed immunity, which causes malfunctioning in all body cells, decreasing the body’s defense against multiple pathologic conditions [50]. The literature reports that higher levels of AF present associations with abdominal adiposity [15], and it was shown to be inversely correlated with the obesity prevalence and cardiometabolic risk in European teenage (however, the Portuguese cities were excluded) [51]. Moreover, lower AF significantly affects physical function across all age groups, including an increased risk of metabolic and cardiovascular illness disease, and adequate aerobic fitness throughout a lifetime has an inverse relationship with many causes of death [52].

To state the AF importance to health, we wish to highlight the study of Henriksson et al. [53], which investigated the association between a low physical fitness index and cardiovascular disease for a cohort of 1 million adolescents aged 16 years old during the years of 1972 to 1994. Ultimately, they found that cardiorespiratory fitness was strongly associated with cardiovascular disease risk from the 28 year follow-up. In a systematic review and meta-analysis of longitudinal studies, García-Hermoso et al. [54] identified that the AF improvement through a one-year-course potentially protected against cardiovascular disease onset in later life. In addition, similar to our study, Joensuu et al. [55] performed a 2-year observational study with 633 adolescents aged 12.4 ± 1.3 years (50% girls) and, through an ML model, they noted that AF measured by the 20-m shuttle run test predicted obesity and physical fitness perception with a sensitivity of 80% and 60%, and specificity 78% and 79% in girls and boys, respectively.

The lower speed in ST (higher time) and lower HJ are indicators of poor lower limb muscle power and are related to overweight and obesity [56]. Moreover, speed is related to lower limb power and sprinting capacity, also leading to relationships with lower limb strength, which positively correlates with obesity [15], metabolic syndrome, and cardiovascular disease in children and adolescents [57]. Additionally, maintaining adequate lower limb power levels throughout early life and adolescence results in good physical functionality in daily living activities, which reduces sedentary behavior [58]. In an observational and cross-sectional study, Petrovics et al. [59] analyzed a cohort of 708 adolescents (360 girls) aged 14–18 years old, and during a 4-year follow-up, they found worsened lower limb power in the obese and overweight boys and girls. Furthermore, in a systematic review, Méndez-Hernández [60] found significant and positive effects of resistance training in reducing body fat percentage in children and adolescents, highlighting the importance of developing muscle strength as a safe and indirect way to prevent and reduce obesity rates in childhood and adolescence [60].

We found negative associations between ULS and obesity, indicating that higher strength values are related to reducing the adolescent’s obesity risk. However, this variable should be interpreted cautiously; the literature reports lower associations or no differences between obese and non-obese youths’ upper limb strength [61]. In this way, the physical functionality may differ, whereas the non-obese trend could present with no higher functionality scores [62]. In contrast, our NNET interpreted the inverse associations between lower limb muscle power and obesity with an accuracy of 75%, 100% recall, finding true and negative obesity cases, with a K-Folds CV ROC AUC of 66%, showing that at least in our study, the weaker adolescents were more obese. ULS is related to good posture, better bone health, perceived well-being, and self-esteem during adolescence [63]. Moreover, ULS is inversely related with cardiovascular risk in teenagers [64]. Furthermore, establishing a good strength foundation during adolescence is a significant factor for long-term health, in adulthood and later life [65].

Similar to our results, in a cross-sectional study in Texas, USA, Ajisafe [66] analyzed data from 210 children aged 9.7  ±  1.08 years and he found an inverse association between ULS and cardiorespiratory fitness in the participants. Further, Ajisafe also found an inverse association between ULS and the participant’s body weight [66]. Agreeing with our results, with data from the same study cited before, Petrovics et al. [59] performed a 4-year follow up study with 708 adolescents (360 girls) aged between 14–18 years and they also found an inverse association between the ULS and weight status of the adolescents at the end of the fourth year.

We identified an inverse and small correlation between lower limb flexibility the obesity in adolescents. Flexibility is directly related to a good joint range of motion, which is essential for keeping functionality in all phases of life [67,68]. During adolescence, flexibility plays a crucial role in obesity risk in some respects. Flexibility is essential to good sports performance and adherence, allowing teenagers to develop themselves better in sports practice [69]. Due to successful sports practice, adolescents can develop all health-related physical capacities, i.e., muscle strength/power and cardiorespiratory fitness [70]. On the other hand, flexibility could not always be correlated to obesity [71]. Flexibility can be determined by several factors, such as genetics, sociodemographic, physical activity, and exercise type practiced [68]. Therefore, individuals with different body structures could present a non-linear trend in their flexibility levels, making this variable not the best obesity-associated factor [72].

Conversely, poor flexibility negatively influences the adolescent’s global range of motion, making the sportive practice uncomfortable and demanding, thus, reducing their sport participation and increasing their sedentary behavior [73]. As an example of this, in our study, flexibility contributed little to the adolescent’s obesity risk, showing that in our study, flexibility was not the major influencing factor to their obesity risk. In addition to these evidence, Bataweel and Ibrahimb [74] performed a cross-sectional study with 90 scholar children and adolescents aged 6–11 years old, and they found that LLF was lower in the obese participants. Apparently, obesity also exerts an inverse influence on obesity, which can be seen in the study of Molina-García et al. [71], that in a cross-sectional study with 196 children aged 5–10 years. As result, this study verified that the obesity condition reduced the foot joint flexibility in the participants. This evidence states that, possibly, a negative and intercommunicating cycle between obesity and flexibility [71].

Finally, the adolescent’s age and sex also had small correlations with their obesity risk. Chronological age and sex are complex processes of adolescence. During adolescence, rapid psychological and hormonal changes happen with the purpose to prepare all body systems for adulthood and the rest of life. However, this accelerated process makes some individuals more susceptible to becoming obese [30]. This fact is more pronounced in girls than boys, due to female hormone production being more associated to greater body fat increases [75]. In addition, adolescence is a turbulent time of life, where the adolescent’s psychological aspects could be influenced negatively by their elevated emotional stress, possible eating disorders, self-stigmatization with their body image, and in most severe cases, depression and other mental illness which could affect the obesity incidence during adolescence [76].

Indeed, behavioral factors are considered strong modulators of the adolescent’s overall health; as a further instance of this, Pojskic and Eslami [77], in a cross-sectional study with 753 Croatian adolescents aged 10–14 years (361 girls), verified that sex was not associated with higher abdominal obesity, but in really, the adolescent’s physical activity levels were the most impacting factor. These results from Pojskic and Eslami [77] agreed with our results, which showed that a healthy lifestyle is noted as a strong protective factor for obesity in childhood and adolescence, independently of sex.

### Study Limitations and Perspectives

As limitations, we believe that our dataset (654 participants) cannot perfectly express the obesity rates among adolescents, which can change when analyzing greater datasets. In addition, our dataset presented 10% missing data (65 participants), so we took two actions. First, we ran all algorithms, excluding the missing rows, and second, we considered them but replaced them with the mean of the column. Despite the ideal model being free from missing values from the dataset, it is rarely observed in real practice, even more so when it comes to population data. Conversely, we did not pose performance with or without the two dataset conditions, so it was not considered a deficiency of a robust study. Other artificial intelligence techniques related to obesity prediction are presented in the literature with samples between 498 and 2.1 billion participants, showing the AI applicability, independent the dataset’s size [78]. Indeed, we believe that the lack of other obesity-related independent variables beyond physical fitness did not allow us to obtain higher accuracy. We failed to find studies with NNET to assess the obesity risk linked to the physical fitness of Portuguese adolescents, which does allow us to highlight our study as the first investigation applying an NEET to assess the obesity risk of Portuguese adolescents in relation to their physical fitness levels. Therefore, our NNET presented good accuracy, precision, and validation, based on low-cost measuring variables, which bring a good generalization performance; thus, health professionals can strongly trust in this information for their real practice and act efficiently against the obesity pandemic. Despite the good performance of our NNET, we encourage researchers to replicate this NNET in bigger datasets in Portugal and even in another countries, and consider additional influencing factors such as environment, genetics, pubertal transition, diet, lifestyle, physical activity, medication, and psychological status. We believe that these actions may help researchers to produce a wider viewpoint about obesity incidence in relation to the physical fitness levels of adolescents in Portugal.

## 5. Conclusions

In summary, our NNET presented good accuracy (75%), and K-Folds CV validated it with similar results [good accuracy (71%) and ROC AUC (66%)]. According to our NNET, there was an increased obesity risk when linked to low physical fitness in Portuguese teenagers. In addition, we found moderate effect size correlations between most of the physical fitness-related independent variables (AF, ULS, and ST) with the adolescent’s BMI percentiles, indicating an increased obesity risk linked to low physical fitness in adolescents. Our study’s results had good generalization and must be taken as a basis for applied practice for Portuguese health professionals, which could use these low-cost variables to screen for less-fit adolescents and introduce them to early sports practice. In addition, our results should allow for the mitigation of adolescent obesity risk through simple lifestyle changes. Therefore, this must be considered for national health policies that could use our results to implement long-term government programs to keep teens physically active and mitigate the obesity epidemic in Portugal.

## Figures and Tables

**Figure 1 behavsci-13-00522-f001:**
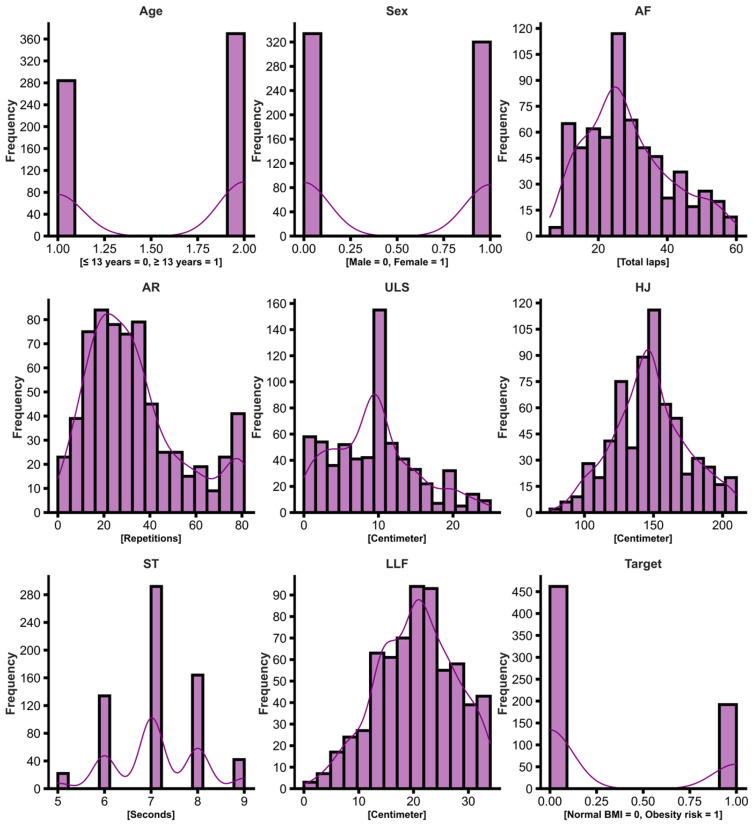
Histogram showing the frequency of subjects in each independent variable, Age, Sex, AF, AR, ULS, HJ, ST, LLF, and the Target variable (obesity risk classification according to BMI percentiles, 0 = negative, 1 = positive).

**Figure 2 behavsci-13-00522-f002:**
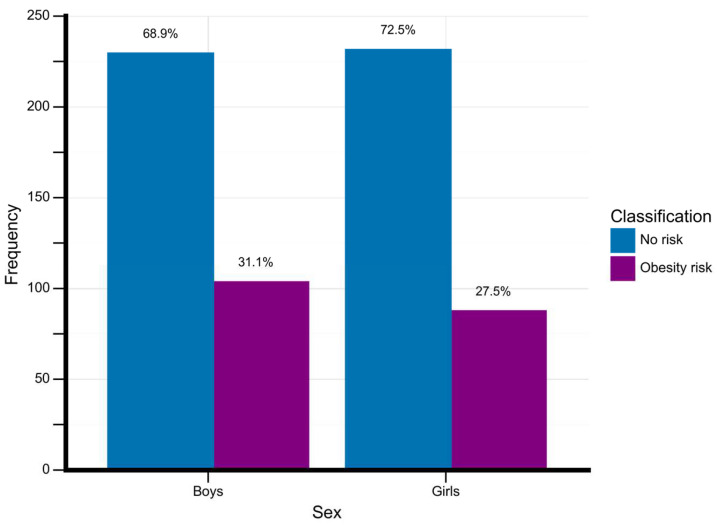
Obesity rates by sex. Data is described in absolute and percentage. Frequency, total number of adolescents for each classificatory group.

**Figure 3 behavsci-13-00522-f003:**
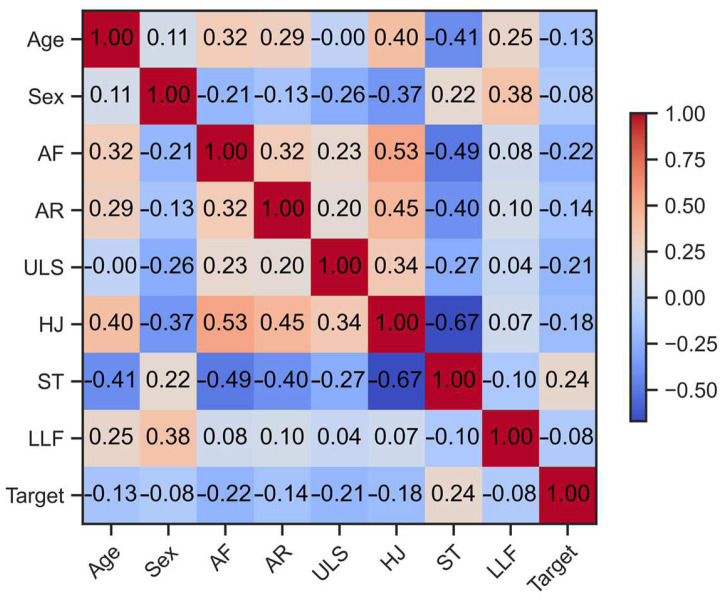
Correlation matrix of explicative variables Age, Sex, AF, AR ULS, HJ, ST, LLF, and the Target variable (obesity classification according to BMI percentile values).

**Figure 4 behavsci-13-00522-f004:**
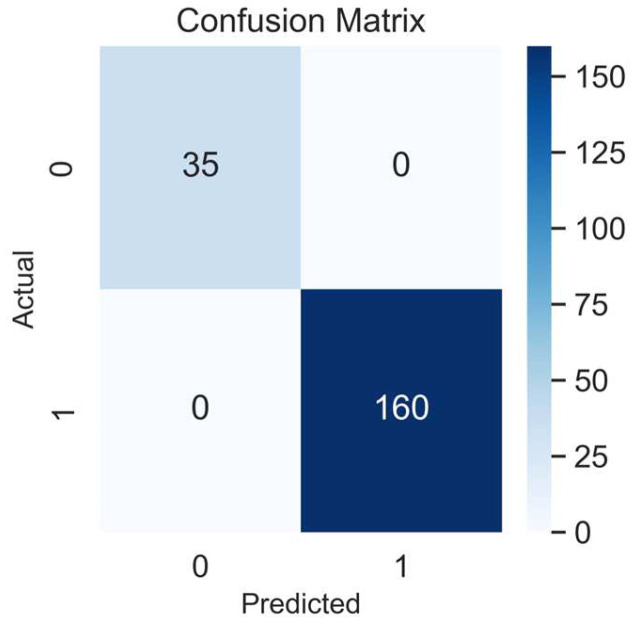
Confusion matrix of the NNET model. 1, positives, 0, negatives.

**Figure 5 behavsci-13-00522-f005:**
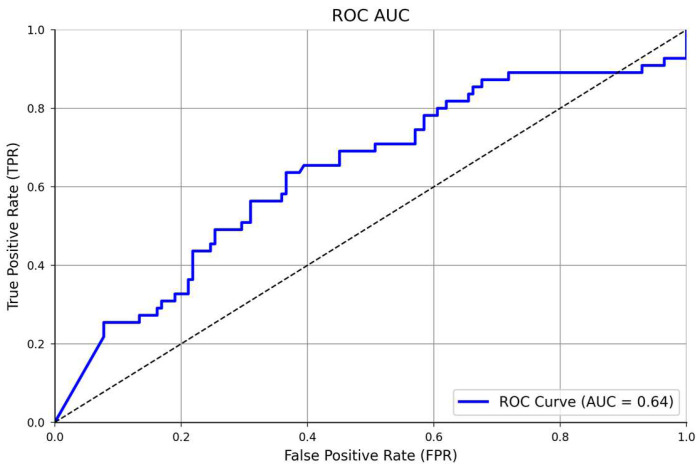
ROC AUC analysis of the classificatory NNET’s performance.

**Figure 6 behavsci-13-00522-f006:**
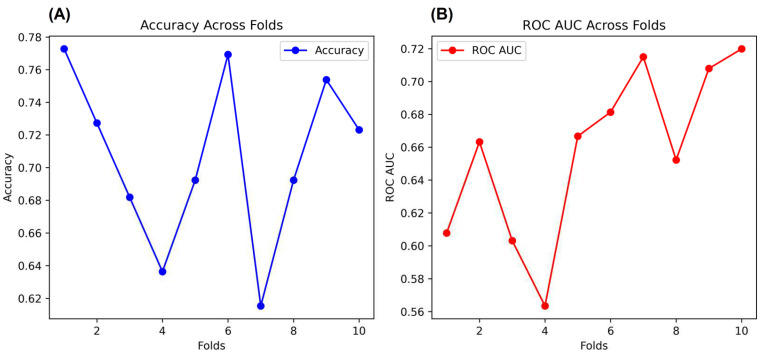
Cross-validation of the NNET. Figures (**A**,**B**) show the accuracy and ROC AUC, respectively, across the folds.

**Table 1 behavsci-13-00522-t001:** Anthropometric and functional characteristics of the participants by sex.

Variables	Boys ($\bar{X}$ ± SD)	Girls ($\bar{X}$ ± SD)
Age (Years)	14.6 ± 2.2	13.01 ± 2.06
Weight	54.45 ± 15.75	58.51 ± 15.49
Height	158.37 ± 11.60	167.4 ± 79.8
AF (Laps)	31 ± 13.4	25.7 ± 10.12
ULS (Repetition)	10.86 ± 5.86	7.87 ± 5.08
HJ (Centimeter)	156.84 ± 27.23	137.89 ± 22.22
40-m ST (Seconds)	6.82 ± 0.95	7.41 ± 0.75
LLF (Centimeter)	18.29 ± 6.88	22.67 ± 6.62

Note: $\bar{X}$, mean; SD, standard deviation.

**Table 2 behavsci-13-00522-t002:** Parameters structure of the NNET.

classifier = Sequential()
classifier.add(Dense(activation = “relu”, input_dim = 7,
units = 4, kernel_initializer = “uniform”))
classifier.add(Dense(activation = “relu”, units = 8,
kernel_initializer = “uniform”))
classifier.add(Dense(activation = “sigmoid”, units = 1,
kernel_initializer = “uniform”))
classifier.compile(optimizer = ‘adam’, loss = ‘binary_crossentropy’,
metrics = [‘accuracy’])

**Table 3 behavsci-13-00522-t003:** Summary of the NNET model.

Layer (Type)	Output Shape	Parameters
dense (Dense)	(None, 4)	32
dense_1 (Dense)	(None, 8)	40
dense_2 (Dense)	(None, 1)	9
Total Parameters: 81		
Trainable Parameters: 81		
Non-Trainable Parameters: 0		

**Table 4 behavsci-13-00522-t004:** Accuracy outputs for the epochs units used in the NNET.

Epochs	Processing Time	Accuracy Loss	Accuracy
1/24	0 s 3 ms/step	0.5188	0.7505
2/24	0 s 4 ms/step	0.5188	0.7462
3/24	0 s 3 ms/step	0.5191	0.7505
4/24	0 s 4 ms/step	0.5192	0.7527
5/24	0 s 3 ms/step	0.5187	0.7484
6/24	0 s 3 ms/step	0.5191	0.7418
7/24	0 s 3 ms/step	0.5186	0.7505
8/24	0 s 3 ms/step	0.5190	0.7505
9/24	0 s 2 ms/step	0.5186	0.7549
10/24	0 s 3 ms/step	0.5189	0.7462
11/24	0 s 3 ms/step	0.5185	0.7440
12/24	0 s 3 ms/step	0.5155	0.7502
13/24	0 s 2 ms/step	0.5187	0.7505
14/24	0 s 2 ms/step	0.5190	0.7505
15/24	0 s 3 ms/step	0.5182	0.7484
16/24	0 s 2 ms/step	0.5186	0.7505
17/24	0 s 2 ms/step	0.5185	0.7527
18/24	0 s 3 ms/step	0.5187	0.7484
19/24	0 s 3 ms/step	0.5184	0.7505
20/24	0 s 4 ms/step	0.5187	0.7484
21/24	0 s 4 ms/step	0.5181	0.7484
22/24	0 s 4 ms/step	0.5185	0.7484
23/24	0 s 3 ms/step	0.5183	0.7527
24/24	0 s 4 ms/step	0.5184	0.7527

## Data Availability

Data are available upon request from the corresponding author.

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
