# Peer review of "A Deep Learning Neural Network to Classify Obesity Risk in Portuguese Adolescents Based on Physical Fitness Levels and Body Mass Index Percentiles: Insights for National Health Policies"

_behavsci, 2023, doi:10.3390/bs13070522_

Round 1

Reviewer 1 Report

The authors must be commended for carrying out a study regarding the development of an artificial neural network model that identifies the risk of obesity in adolescents based on their body mass index percentiles and levels of physical fitness. This topic is important, interesting, and relatively novel. However, some issues need to be taken into consideration.

Introduction

Line 53: New paragraph.

Line 68: New paragraph.

Line 71-79: I think this part belongs to the third paragraph, where you mentioned again the Fitescola program (Line 80-88).

Line 115: ‘’In addition, we published preliminary results about the relationships between lower physical fitness and Portuguese adolescent’s obesity...’’. Please add a reference.

Please add a hypothesis.

Results

I suggest adding a table with descriptive values of measured parameters.

Methods

Line 128 and 129: ’’This observational, cross-sectional study aims to classify the obesity risk in Portuguese adolescents from both genders, based on body mass index (BMI) percentile values.’’ – What about fitness level?

You did not provide information about the fitness level assessment. Please add these informations.

Figure 1 is quite unclear, from my point of view. I suggest better describing it in the legend or in the text.

I think Figure 3 belongs to the results section.

I generally suggest expanding the method section, especially the measurement part. I can see you provided a reference about it (17), but I think it would be more appropriate if the future publication contains these informations.

Discussion

Generally, I think you should expand the discussion section and explain in more detail the observed results regarding the relationship between physical fitness tests and obesity. For example, you did not explain the relationship between lower limb flexibility and obesity…

Author Response

Dear Reviewer,

Thank you very much for the time you spent and your feedback on this manuscript. We have made every effort to take on board your recommendations and comments. We hope this revised second version and the responses to the comments (kindly refer to our replies below) will meet your requirements. Please, note that all new changes in the revised manuscript are edited with the Microsoft Word® tracking tool. Please, check the point-by-point answer.

Please, see attachment.

With best regards,

José Eduardo Teixeira.

Reviewer 2 Report

This manuscript aims to develop an artificial neural network model (NNET) that identifies obesity risk in Portuguese adolescents based on their body mass index (BMI) percentiles and fitness levels. Obtaining 75% accuracy in the model. 

Congratulations for this model, but I have some doubts to solve. 

-As a conclusion, an increase in obesity is linked to low fitness. This has been known for a long time. What is new in this model?

How could they increase the 75% accuracy in the model? In the end more than 2 out of 10 people will not be recognised with this model. 

What is the value of this model or what is different about it that could be useful for example in a school to detect obesity so that teachers can do something about it?

What would the transfer of this model to society look like?

Regarding the format, figure 1 and figure 2 could be clearer and with better quality. 

Author Response

(The authors gave the same response as above.)

Reviewer 3 Report

The authors aimed to develop an artificial neural network (NNET) model that identifies the risk of obesity in Portuguese adolescents based on their body mass index (BMI) percentiles and levels of physical fitness. The NNET had good accuracy in identifying the risk of obesity in Portuguese adolescents based on the BMI percentiles. Correlations of moderate effect size were perceived for aerobic fitness, upper limbs strength, sprint time, lower limbs flexibility, and horizontal jump, showing that physical fitness levels contributed to the obesity risk. According to NNET, there was an increased risk of obesity linked to low physical fitness in Portuguese teenagers. I read this article with great interest, and I have a few comments, hope can improve the manuscript.

1.     In Figure 1, please replace 0 and 1 by Boys and Girls in the graph to make it straightforward.

2.     In Figure 4, code is suggested to be input as text, rather that screenshot.

3.     The authors explain that they used a data set of 654 Portuguese adolescents. I do not see any discussion on missing data issue in the manuscript, how did the author handle the missing data?  Can you also discuss the impact of it on the conclusions? My concern is that missing data issue may exist and bias the results.

4.     The authors used data from only Portuguese adolescents. However, it would be beneficial to repeat the study with data from other countries or regions to see if the results are similar. 

5.     There are many analysis models existing to answer this question, but the author only uses NNET model to draw the conclusion. I would suggest that authors try other different models (at least one) as supplementary analysis (results could be in the Appendix) to compare if the conclusions are robust. 

Author Response

(The authors gave the same response as above.)

Round 2

Reviewer 1 Report

Dear Authors,

Thank you for taking into consideration my comments and suggestions. I believe the manuscript is now suitable for publication. Best regards.

Reviewer 3 Report

Thanks much for revision.  I do not have further comment.  Good luck!